# The Influence of Nurse-Led Interventions on Diseases Management in Patients with Diabetes Mellitus: A Narrative Review

**DOI:** 10.3390/healthcare12030352

**Published:** 2024-01-30

**Authors:** Hamad Ghaleb Dailah

**Affiliations:** Research and Scientific Studies Unit, College of Nursing, Jazan University, Jazan 45142, Saudi Arabia; hdaelh@jazanu.edu.sa

**Keywords:** diabetes mellitus, chronic patients, nurse-led interventions, self-management, patient outcomes

## Abstract

The global prevalence of people with diabetes mellitus (PWD) is rapidly increasing. Nurses can provide diabetes care for PWD in several areas. Interventions led by nurses can support PWD for effective management of diabetes, which can positively improve clinical outcomes. Nurse-led diabetes self-management education (DSME) is an effective strategy to manage diabetes mellitus (DM) since it improves self-care practice and knowledge regarding diabetes. PWD often need to stay in hospitals longer, which involves poorer patient satisfaction and clinical outcomes. Nurse-led clinics for DM management are a new strategy to possibly ameliorate the disease management. Diabetes specialist nurses can play an important role in improving diabetes care in inpatient settings. Various studies have revealed that nurses can independently provide care to PWD in collaboration with various other healthcare providers. Studies also demonstrated that the nurse-led education-receiving group showed a significantly reduced level of average glycosylated haemoglobin A1c level. Moreover, nurse-led interventions often result in significant improvements in diabetes knowledge, psychological outcomes, self-management behaviours, and physiological outcomes. The purpose of this literature review was to identify the impact of nurse-led interventions on diabetes management. Moreover, in this review, a number of nursing interventions and the nurses’ roles as educators, motivators as well as caregivers in DM management have been extensively discussed. This article also summarises the outcomes that are measured to evaluate the impact of nursing interventions and the strategies to overcome the existing and emerging challenges for nurses in diabetes care.

## 1. Introduction

Diabetes mellitus (DM) is a chronic, metabolic disease caused by defects in insulin function, insulin discharge, or both, which eventually results in increased blood glucose levels [1]. Globally, DM is one of the top four non-communicable diseases and the most common metabolic disorder. The major DM types include type 1 DM (T1DM) and type 2 DM (T2DM), where T2DM is more prevalent compared to type 1 and accounts for 90–95% of all DM cases [2,3]. DM is considered as a main public health concern globally due to the steady rise in the number of people with DM (PWD) [4,5]. It has been reported by the World Health Organisation (WHO) that globally, the number of PWD has significantly increased from 108 million in 1980 to 422 million in 2014. Moreover, this number will continue to increase to 592 million by 2025. In the adult population, the worldwide occurrence of DM has almost doubled since 1980, increasing from 4.7% to 8.5%. Globally, DM-associated deaths also markedly rose by 70% between 2000 and 2019. DM is rapidly increasing in developing countries wherein around 77% of PWD live in middle- and low-income countries [3,4]. According to the International Diabetes Federation, DM cases will continue to increase and will reach 9.9% by 2030 worldwide [6]. PWD are at greater risk of complications and more prone towards a longer length of stay (LOS) in hospital [7]. In 2019, the National Diabetes Inpatient Audit (NaDIA) reported that 18% of all hospital beds were occupied by PWD, which had risen from 14% in 2010 [8]. Therefore, DM requires attention since DM-related complications can result in poor health outcomes and overuse of healthcare resources and services.

In most cases, DM is a secondary reason instead of the primary reason for hospital admission, thus individuals are more commonly under the care of non-diabetes specialists. Furthermore, it has been reported that the doctors in training in these teams often lack extensive knowledge in managing DM, where only 28% were found to be fully confident in DM management [9]. Along with other specialties, specialist nurses and nursing teams are increasingly becoming involved in providing care to PWD. These nurses deliver education and support for both patients and staff across specialities and give clinic or phone contact to facilitate the discharge of patients or avoid hospital admission in a timely manner. Nurses are in a better position to provide care and education to PWD compared to other healthcare professionals since nurses spend most of their time with patients. In addition, nurses are also in a better position to provide measures and best care practices to patients regarding diabetes management compared to other healthcare professionals including doctors [10]. In a study, Lou et al. [11] concluded that nurses are typically better listeners and possess better knowledge of PWD compared to other healthcare professionals. Collectively, these findings suggest that the commitment and attitude of nurses to the care of PWD are usually higher compared to other healthcare professionals [10]. Diabetes inpatient specialist nurses (DISNs) are highly skilled nurses who can coordinate, educate, counsel, motivate, lead, and help in the care management of PWD in diabetes care. The National Health Service (NHS) also reported the significance of DISNs. Moreover, the National Institute of Clinical Excellence (NICE) recommended that there should be at least one DISN per 300 hospital beds [12]. However, still there is a lack of dedicated DISNs in clinical settings. In 2018, the NaDIA reported that there are still no dedicated DISNs in 22% of hospitals [12].

The growing rate of T2DM indicates that there is a need for new models of practice and care for T2DM management. Various studies have also suggested the necessity to prepare nurses with the required knowledge to facilitate best practices for PWD. There is also ambiguity regarding the roles of nurses in providing PWD, which is predominantly seen in countries including Saudi Arabia that do not have DISNs as a recognised group. Moreover, although nurses contribute significantly to efficient DM management, their collaborative contributions with other healthcare providers are less recognised [13]. Nevertheless, healthcare systems are increasingly embracing nurse-led models that are supposed to be more patient-centred, as opposed to the traditional physician-led models that reflect a medically-oriented model of care. It has been revealed that with proper training, nurses can efficiently play roles in the management of DM, and present trends have observed a change in the tasks executed by nurses, which were formerly provided by physicians [13]. Furthermore, nurses can develop, implement, and ensure effective DM interventions by directly providing care delivery, supervising care delivery, and training non-medical personnel in providing DM care for a range of patients [14]. This review article summarises a number of nursing interventions and roles as educators and motivators as well as caregivers that play important roles in T2DM management. In addition, a range of outcomes that are measured to evaluate the impact of nursing interventions and the strategies to overcome the existing and emerging challenges for nurses in diabetes care are extensively discussed.

## 2. Search Methods

A range of widely used and popular databases including the Cochrane Database, Google Scholar, PsycINFO, PubMed, Web of Science, Scopus, ScienceDirect, and CINAHL were searched from 2010 to October 2023. The terms that were used to search the databases included “nursing”, diabetes, “intervention”, “diabetes self-management”, “type 2 diabetes”, “nurse-led diabetes management”, “diabetes inpatient specialist nurses”, “nursing AND diabetes”, “nursing AND diabetes AND intervention”, and “nursing AND diabetes self-management”. The search method for this review aimed at identifying all review articles, original research, clinical trials, and books that contained information regarding the role of nurses in the management and prevention of DM. References from the selected literature have been cited where relevant.

## 3. Roles of Nurses in the Management of Diabetes

### 3.1. Educating Patients with Diabetes

Increasingly, diabetes nurse educators (DNEs) contribute significantly in equipping patients with confidence and knowledge to attain self-care goals for DM management [15]. There are seven major factors in effectual self-care management involving healthy coping strategies such as reducing risks, problem-solving, glucose level monitoring, medication adherence, exercise, and a healthy diet [16]. However, the effectiveness of health education activities depends on a patient’s acceptance of their diabetes. Additionally, sociodemographic factors such as an individual’s education level can impact adherence to self-care in DM management. Care delivery techniques also need to be carefully considered. The use of pictures and teach-back techniques are suggested techniques for individuals with low literacy [17]. On the other hand, one-to-one consultation was found to be more effective compared to group-based consultation [18,19]. Several studies have already evaluated the impact of nursing care and have observed that nurses play an important role in educating patients to manage their disease [20,21]. Additional studies have revealed that nurse-led education has a positive impact on the condition of patients including improved glycaemic controls [21,22].

In a recent study, Bostrom et al. [20] reported the significance of the roles of nurses in patient education, where one of the roles of DSNs was teaching, and they were associated with educating individuals regarding DM, the outcomes of diagnostic tests, and probable complications. In another study, Wexler et al. [21] carried out a randomised trial study with two groups in order to reveal the significance of diabetes education. In that study, the usual care was received by one group, while the other group received both formal education and intervention care from specialist nurses. Participants of the usual care received any care received from diabetes education and care provided by non-specialist nurses and physicians. It was reported in that study that compared to the usual care group, the inpatients in the intervention group had much lower mean glucose levels. On the other hand, the intervention group experienced a reduced level of average glycosylated haemoglobin A1c (HbA1c) in the year after discharge. The HbA1c test is commonly carried out to measure the average blood glucose levels in the past 3 months. Raballo et al. [22] carried out a different study where patients received either group care or the usual care, where the group care led by one to two health operators (including nurse, educator, dietitian, doctor, or psychopedagogue) experienced better positive outcomes. Furthermore, patients who received group care exhibited more positive attitudes compared to those who had traditional visits. Furthermore, individuals in group care expressed a more articulated and wider range of concepts linked with the care they received compared to those who received the usual care and who mostly expressed concepts with negative implications. The findings indicate that individuals under the usual care expressed their setting and condition of care with notions that generally indicated an external locus of control, poor empowerment, and negative attitudes. Collectively, these findings suggest the importance and changing role of nurses in diabetes education in ameliorating glycaemic controls [23]. A summary of the various studies that have been conducted to assess the impact of nurse-led diabetes self-management education (DSME) is outlined in Table 1.

### 3.2. Nurse-Led Diabetes Self-Management Education and Support (DSMES) and Educational Intervention

It has already been shown that self-care knowledge and practice can be improved via DM self-management support and education for patients. In addition, the American Diabetes association (ADA) has also recognised the importance of DSME for PWD and suggested the inclusion of nurse-led DSMES in the management of DM [18]. The International Diabetes Federation, Centres for Disease Control and Prevention as well as the World Health Organisation have also suggested the inclusion of nurse-led DSMES in DM management. As part of routine care, the major goals of DSMES that are monitored and measured include quality of life (QoL), health status, and providing support for self-management and improvement in clinical outcomes [3,31]. In a study conducted in an American Diabetes Association centre, Brunisholz et al. [32] demonstrated that DSME provided by certified diabetes educators including a registered nurse or dietitian could significantly ameliorate the various components of diabetes care bundle including HbA_1c_ level (below 8.0%), low-density lipoprotein level (below 100 mg/dL), blood pressure (below 140/90 mmHg), nephropathy screening or prescription of angiotensin receptor or angiotensin converting enzyme blocker, and retinal eye exam. These findings demonstrate the benefits of adding DSME programmes in the treatment of PWD. Moreover, such programmes involve a low-operating cost [32]. In addition, various techniques have also been established and applied by nurses in a range of settings. Sherifali et al. [33] evaluated the impact of a patient-tailored and computer-generated feedback intervention on glycaemic control [33]. The impact of a hospital-based clinic intervention was assessed to ameliorate glycaemic control in a different study [34]. In another study, Kang et al. [35] compared the efficacy of conventional care with the family partnership intervention in terms of DM management [36]. Typically, nurse-led studies also consider socially appropriate DM interventions [37].

Various studies have also explored the influence of nurse-led educational intervention. A study was carried out with newly conducted focus groups for individuals with T2DM to detect educational needs and establish an educational intervention based on the focus group findings [38]. A symptom-based educational intervention was also developed to address DM symptoms and allow patients to select self-management approaches for symptoms based on unique preferences and needs. Furthermore, a web-based DSME programme was also developed that allowed patients to continue with modules at their own pace and achieve target goals. In general, educational intervention studies have been developed to assess the impact of certain diabetes education programmes on various outcomes such as glycosylated haemoglobin (HbA1c), self-management activities, and diabetes knowledge [38,39]. On the other hand, other studies have assessed the impact of novel educational methods such as telephone-delivered education, multimedia educational programmes, web-based diabetes education, and tailored educational programmes [14,40]. Collectively, the findings from all of these studies indicate the positive impact of the aforementioned educational methods.

### 3.3. Interprofessional Teamwork in Diabetes Care

Nurses, as a member of an interprofessional healthcare team, adopt various methods to help with the self-management of PWD (Figure 1). Nutritionists and nurses often work as a team to provide support to individuals with proper intake and food choices, which is considered as one of the most challenging areas of DM self-management [41,42,43]. Furthermore, nurses work in association with psychologists to counsel PWD [44]. Nurses also closely work with physicians to screen patients and change healthcare plans as required. The ADA suggests that PWD ought to receive DSME as per the National Standards for DSMES at diagnosis and as required afterwards. Both nurse-led groups and individual sessions are conducted to provide diabetes management education [34,37,39,41,42,44,45]. A web-based format might also be used to deliver education. Moreover, educating partners or family members of PWD is suggested and is involved in nursing education [14].

### 3.4. Nurse Prescribing in Diabetes Care

Nurse-led clinics for DM management are a new strategy to possibly ameliorate the disease management [46,47,48]. These clinics can vary with regard to work delegations and structure. As part of typical DM management, nurses provide education and patient support with a particular focus on insulin administration. Some clinics have recently extended the role of nurses to include prescription and drug therapy monitoring. Nurses work as substitutes in these clinics to complement physicians in DM management. The primary goal of this care delivery model is to allow individuals to gain access to effective and safe healthcare on time. Prescribing medicines includes initiation, changing, or stopping medication dosages. In addition, the role of nurse prescribers can vary from protocols that restrict their roles to specify very specific indications and selective drugs to comparative independence in terms of all the aspects of prescribing including drug selection. Both training and regulatory environments are pertinent to the framework adopted in any specific setting [49]. For example, in New Zealand, registered nurses practising in specialty teams and primary health are authorised by the Nursing Council of New Zealand to prescribe certain medicines including antidiabetic drugs following the completion of a Nursing Council of New Zealand-approved postgraduate diploma.

Various studies have confirmed the beneficial outcomes of the role of nurses in ameliorating DM management [50,51]. Nonetheless, such studies do not emphasise nurse prescribers, but rather explain the positive roles of nurses in chronic disease management. In 2010, a meta-analysis was conducted to examine the impact of nurse case management interventions on glycaemic control, which revealed a clinically significant control of blood glucose level as measured by HbA1c. In another study, it was reported that nurse-led interventions, by means of structured algorithms for care, were linked with decreased levels of cardiovascular disease (CVD) risk factors including high blood pressure in diabetes. Martinez-Gonzalez et al. [51] revealed that there was no marked difference between physician-led care and nurse-led care in decreasing HbA1c levels.

### 3.5. Education and Support for Community Health Advisors

Lay health workers or community health advisors can provide diabetes care and education. Nurses can provide support and education to community health advisors. In order to assess the possibility of utilising promotoras (community health workers) to provide a self-management intervention of DM, nurses in a study provided training and education to promotoras for 8 weeks. The nurse and promotora provided a diabetes educational course after the completion of training. After the course, promotoras visited homes to present tailored educational sessions with the nurses, who were accessible for consultation as required [37]. Along with community health advisors, telecarers can also be trained and educated to deliver self-management support to PWD. Nurses have the capacity to supervise and educate telecarers, who can telephone participants about blood glucose control, medication adherence, and knowledge regarding DM. A study compared the printed version of diabetes self-management interventions and telephone conversations. In that study, health educators were supervised and trained by a diabetes nurse educator. In addition, the telephone conversations of the health educators mainly focused on the adherence to medications that were used to treat DM, however, those conversations also focused on lifestyle changes via physical activity and healthy eating [14,40].

### 3.6. Nurse-Led Telephone Interventions

Telephone counselling sessions are used to provide support and education to PWD (Figure 2). In one study, participants were contacted by a registered nurse at a minimum of once a week for a twelve week period. The calls consisted of the self-monitoring of blood glucose, medications, exercise, and a discussion of diet. The study participants were permitted to ask questions and adjust the treatment plan as required, depending on their dietary intake and blood glucose record. In a different study, participants received pre-recorded telephone calls on a weekly basis. Furthermore, participants received an automated health education message and nurses followed up through phone calls according to the responses of the participants during these calls. Individualised care can be provided through telephone counselling to PWD, as confirmed in a study where participants completed questionnaires associated with the self-management of DM. Responses of the study participants were utilised to provide individualised instruction and support as well as guide telephone sessions. Nurses can also monitor and provide education by telephone interventions. In order to manage DM, a telehealth nurse in one study monitored insulin doses and blood glucose levels using the web. Following a review of the patient data, the telehealth nurses called the study participants as required as per the data to suggest a follow-up with the primary healthcare provider or alterations in the treatment plan [52,53].

On the other hand, personalised interventions include the modification of interventions in order to meet individual preferences, abilities, and needs along with nurses to support individuals to follow a suggested treatment plan [54]. If needed, nurses can also deliver customised education plans as per the patient-developed goals and assessment for DM management [35]. Moreover, educational sessions might be changed based on the DM self-management experiences and questions of the participants. Nurses focus on empirical learning in order to solve problems and set targets for the self-management of DM [44]. Nurses can also play a role as coaches for PWD, since they guide them through the self-management treatment plan. Nurses can also help patients as coaches to set personal goals for self-management [55] and assist individuals to problem-solve and modify their goals if patients encounter problems by using their own knowledge regarding DM or with the help of procedures established by the healthcare team [14].

**Figure 2 healthcare-12-00352-f002:**
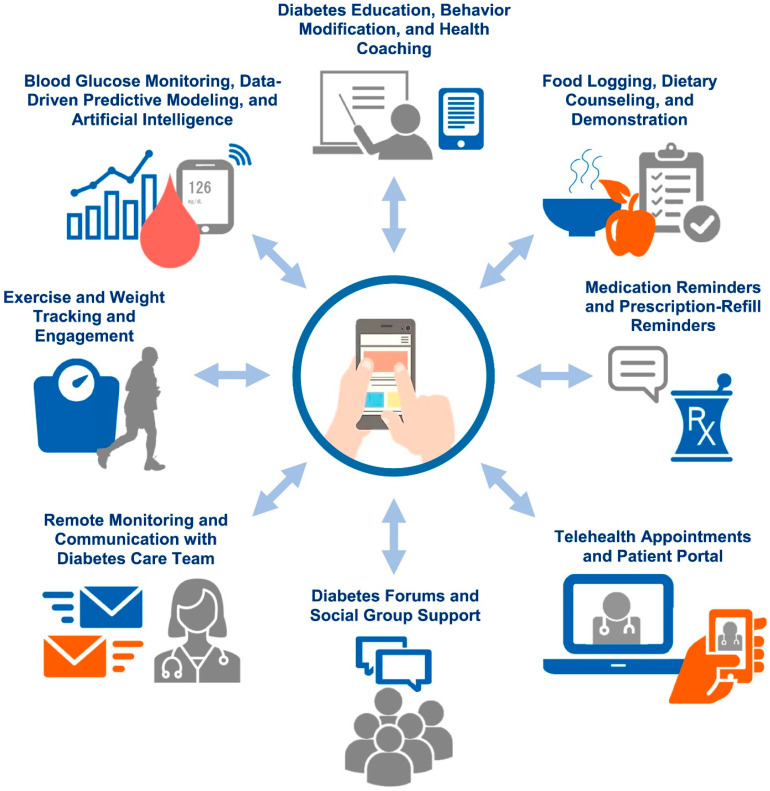
A digital ecosystem of diabetes care is required for effective diabetes management. Reproduced with permission from Elsevier, Reference [56].

### 3.7. Psychological Care and Counselling

In both primary and secondary care, DSNs play crucial roles during hospital stays. In an intervention review, it was revealed that patients preferred to contact their DSNs compared to their GPs. Furthermore, DSNs provide social, psychological, and emotional care for the patients and their family members [57]. It was also observed that patients who remained in contact with DSNs were typically more satisfied with the level of care that they received. Indeed, DSNs exerted a positive impact on the outcomes of diabetes patients. The roles of DSNs are important for both patients and their family members to develop confidence and trust in healthcare providers as well as for optimum health promotion [57]. DSNs also play an important role in reducing communication gaps between patients and clinical partners by playing a role as an intermediate that can avert an escalation of problems [58]. Such communication is particularly important when DM progresses and requires treatment modification such as adding new diabetes medications (for example, insulin) or increasing the dosage of current medications. Fear and anxiety are typically observed in PWD with increased medical complications and poor metabolic outcomes [59]. In one study, a diabetes patient reported feelings of anxiety, fear, and concern while visiting the hospital since the hospital staff did not have adequate knowledge and expertise in DM management. Therefore, it is clear that psychological advice and care is essential for PWD, particularly in inpatient settings [60].

Nurses play an important role in motivating PWD. Various studies have demonstrated the significance of nurses in providing psychological support to PWD. Compared to doctors, nurses are more aware of the needs of patients and also observe the psychosocial problems of nurses that have a significant effect on the control and self-care of PWD. Although nurses provide more psychosocial support and care, they often find themselves less capable of taking care of the psychosocial needs of a patient compared to taking care of their physical needs. Therefore, nurses frequently refer patients to psychosocial specialists. Nurses have also observed that they need to help their diabetes patients feel more hopeful and secure. It was also observed that nurses participated in assisting individuals to address illiteracy and denial. Nurses also use different strategies to encourage patients including humanising complexity, treasuring the relationship, reflecting and advocating on actions, and educating for empowerment [61,62].

### 3.8. Nurses as Advanced Caregivers

As per the definitions for advanced practice nurses (APNs) provided by the Nursing and Midwifery Council and the United Kingdom (UK) regulatory board, skilled nurses have the capacity to exert various roles, for example, ensuring that the care and treatment provided to patients is based on best practice, being a team leader, performing physical examinations as well as providing and deciding treatment [62]. APNs play an integral role in the medical management and education of PWD. Various studies have already mentioned the role of nurses in prescribing medication, where some of the studies showed differences regarding the level to which nurses have that duty. In one study, over two-thirds of specialised nurses in the UK prescribed medicine for common DM complications such as cardiovascular disease, hyperlipidaemia, and hypertension, however, they dedicated no more than 20% of their week in prescribing medications. This finding suggests that nurses spend most of their time in other nursing care activities and less time is dedicated to ordering medications and on advising. Nonetheless, there is no certain guidelines or indications regarding how much time nurses have to devote in prescribing medications. Moreover, although the nurses are receiving more advanced skills and knowledge for the management of medications to treat DM, still there are concerns regarding their actual involvement in that role [62].

Several studies have also emphasised the role of APNs in screening DM complications. Studies have revealed that nurses are associated with screening for complications in the feet and eyes [63], while other studies have reported the role of nurses in briefing doctors regarding problems or complications. As part of their daily routine, nurses are more associated with carrying out administrative responsibilities for diabetes care. It has also been revealed that nurses have managerial responsibilities including being an administrator, bureaucrat, and executive [20]. Nurses also play an important role as a collaborator. Nurses play an important role as a doctor’s assistant, which is an essential part of the profession. Interestingly, it was observed that nurses considered assisting doctors to be more important than providing effective care and spending time with patients to support and educate them. Moreover, it was reported that general practitioners (GPs) often acted based on the assessments of patients carried out by nurses, which suggests that GPs trust the assessments of practice nurses [63]. Moreover, nurses also advise doctors regarding medicines, help doctors in recommending treatment, inform doctors about problems or complications, and play roles as intermediaries between patients and doctors. Nurses also plan and organise diabetes care between themselves, doctors, and other professionals, wherein they emphasised that they shared their mission in DM care with various other professionals [20].

## 4. Impact of Nurse Interventions on Patient Outcomes

### 4.1. Increased Patient Satisfaction

DSNs have a significant contribution in increasing patient satisfaction. In a study with 214 patients in the UK, Courtenay et al. [64] observed enhanced satisfaction with PWD when consulted by prescribing nurses because of the increased consultation time and the establishment of relations between patients and nurses. It was also reported that around 92% of patients found that the care management programme led by DSNs was moderate to tremendously supportive in managing their condition. In a different study, it was revealed that a DSN-led care program for newly diagnosed patients with T2DM was clinically effective with higher levels of patient motivation and satisfaction. In workshops in London aimed at improving the patient experience and DM care, patients with T1DM mentioned that they would appreciate more support and education for friends and family. Patients with T2DM would like to see more personalised care and also be seen by the same person. This finding indicates that DSNs can ameliorate patient satisfaction via self-management empowerment, educative sessions, and more personalised and longer consultations [60].

### 4.2. Prevention of Hospital Admissions and Shorter Length of Hospital Stay

Indeed, care provided by diabetes inpatient specialist nurses (DISNs) can lead to a reduced length of hospital stay for PWD. It was reported by NHS England that a DISN (1 nurse per 250 inpatient beds) can decrease the length of stay (LOS) for inpatients with DM, and this finding is supported by various studies. A significant reduction in LOS was reported after the introduction of a ward-based diabetes nurse advisor. Pre- and post-intervention practice data collected by nurses suggested that the presence of a DSN prescriber resulted in a median LOS of PWD of 3 days, which led to significant cost savings. A team of DISNs can also ensure appropriate and timely follow-up and discharge. PWD also reported that hospital admissions could be averted by involving DISNs in accident and emergency (A&E) departments. In that study, which was conducted over 3.5 years, the authors reported that a substantial number of people attending A&E received treatment and were discharged home without admission into the hospital. Interestingly, a cost of around GBP 35,000 was reduced at the hospital over 3.5 years by providing patient-focused care and reducing bed occupancy. In a different study, it was demonstrated that fewer hospital resources for patients were consumed whilst under DSN care, and markedly fewer hospitalisations and emergency room visits were observed for preventable diabetes-associated causes. The introduction of a DISN service also decreased the DM-associated excess bed occupancy in a 6-year study. DISNs play an important role in both the promotion of patient self-management and patient education. Increased knowledge regarding DM and greater awareness can lead to shorter hospitalisations. Moreover, educational programmes led by inpatient diabetes educators are associated with reduced readmissions [14,60].

### 4.3. Enhanced Diabetes Knowledge

In nursing intervention studies, diabetes knowledge is frequently measured to assess the impact of education. A study developed and implemented by nurses that involved 52 contact hours over one year of instructional sessions and support on DM self-management and basics observed substantial rises in diabetes knowledge compared to a wait-listed control group. The impact of interactive multimedia on self-directed learning in the knowledge of PWD utilising a collection of nursing and medical instructions regarding DM logged on compact discs was also studied. Diabetes self-management counselling and a booklet were received by the control group. Compared to the control group, substantial enhancement was observed in diabetes knowledge in the intervention group. In another study, the intervention group received monthly telephone discussions as well as group and individual educational sessions focused on family participation while the control group received the usual care. The knowledge scores of the intervention group were markedly greater than the control group [32]. A study evaluated the effectiveness of ongoing group-based DSME, where nurse-led educational sessions were related to T2DM including metabolic control improvement, physical activity, diet, complications, and basics. Indeed, enhanced knowledge regarding DM was observed in the intervention group [42].

Various studies have also observed marked enhancements in diabetes knowledge in the intervention group after educational interventions [14,39,52]. Education interventions regarding diabetes in different formats for control and experimental groups often observed amelioration in both groups. A video behaviour support intervention involving a brochure on DM self-management in the control group and telephone coaching sessions as well as a workbook provided by a registered nurse in the experimental group was studied. Interestingly, a marked rise was observed in both groups without any difference [55]. In another study, slight increases in diabetes knowledge were observed after providing needs-driven as well as patient-centred information and education sessions in the experimental group and conventional DSME in the control group, however, no difference was observed between the groups [43]. The practicability of diabetes and cardiac self-management programmes was tested in a different pilot study where three educational sessions and a follow-up telephone call were provided by a nurse after one week of discharge, and text messaging was used to address questions linked to self-management one week after telephone contact. Slight improvements were observed in diabetes knowledge after this intervention; however, no difference was observed between the control and experimental groups [14,65].

### 4.4. Reduction in Inpatient Harm

Medication errors are commonly observed in inpatients because of the complex nature of DM. In 2017, the NaDIA revealed that 31% of study participants experienced at least one diabetes medication error while staying in hospital. The errors included both medication management and prescription errors. Furthermore, these errors were much greater than errors in other diseases in hospitals. In 2014, an average medicine administration error rate of 3–8% and a prescribing error rate of 7% were observed in hospitals in the United Kingdom. It has been reported by NHS England that DISNs can decrease inpatient harm by decreasing hypoglycaemic events and medication errors. In 2016, the NaDIA emphasised the necessity for healthcare professionals to have the confidence, experience, and knowledge in handling medications of DM to decrease medication errors. Moreover, the Royal College of Nursing, Trend UK, and Diabetes UK demonstrated that significantly reduced levels of insulin error and therefore reduced LOS were observed with DSNs, particularly in registered nurse prescribers.

In 2017, the NaDIA reported that approximately 1 in 800 inpatients with T2DM and about 1 in 25 inpatients with T1DM developed diabetic ketoacidosis during their stay in hospitals. The aforementioned hospital-acquired emergency conditions are potentially fatal and extremely serious; however, these conditions are preventable and should not develop during hospital admission. Therefore, diabetes specialist teams should have sufficient expertise, capacity, and knowledge to reduce these emergency conditions by means of the personalised and specific management of medications [58]. In one study, the data of 56 PWD were collected at inpatient care over 8 months. The study revealed a positive outcome on medicine delivery systems, and a significantly lower level of medication errors was observed with a DSN study group. Improved glucose levels were also observed in PWD following insulin dose adjustment as per the suggestions of a diabetes nurse educator. Vissarion et al. [57] showed that DSNs have a significant contribution in crisis management. Unfortunately, the number of DSNs is not increasing, despite growing demand for diabetes services. Furthermore, 78% of DSNs shared their concern that their workload was affecting patient care and/or safety.

### 4.5. Self-Management Behaviours

PWD is actively associated with the planning and application of their care. Healthcare professionals ought to help patients with self-management to support PWD to competently and confidently manage DM. Nurses help and assist PWD to problem-solve and set goals for effective DM management. The ADA showed that PWD receive help for self-management behaviours such as monitoring complications, medication administration, self-blood glucose monitoring, physical activity, and healthy eating. Nursing interventions also help patients in changing their behaviour for effective DM management. Nurse-led intervention was found to facilitate healthy behaviours in adult PWD. This patient education programme involved visual aids, problem-solving exercises, discussion, and content presentation. A statistically significant amelioration was observed in health-promoting behaviours including the prevention of complications, hygiene, medication administration, exercise, and dietary behaviour in the experimental group compared to the usual care-receiving control group. In a similar study, a substantial amelioration was observed in glucose self-monitoring, medication adherence, exercise, and diet in adult patients with T2DM in the experimental group [43].

A symptom-focused, nurse-delivered, and in-home counselling and education intervention provided DM intervention modules for patients with T2DM. An extensive improvement was observed in glucose monitoring, diet, and medication practices in the experimental group. However, no significant differences were observed between groups in terms of exercise. A study also evaluated the efficacy of a structured diabetes education programme on self-care. Telephone and face-to-face educational sessions were delivered by nurses addressing problem-solving and self-care. Participation in physical activity and self-blood glucose monitoring were markedly ameliorated in the experimental group, and the alteration was also noteworthy compared to the control group [66]. A psychologist and certified diabetes education nurse cofacilitated sessions that particularly focused on diabetes self-management problem-solving, goal-setting, coping, experiential learning, and questions in a 24-month study. A marked improvement was observed in foot complications, blood glucose monitoring, and diet after 6 months. On the other hand, marked improvements were observed in insulin usage and diet after 24 months [44]. Numerous studies have also reported that nurse interventions can lead to marked improvements in both the control and intervention groups [14,15,33,55].

### 4.6. Physiological Outcomes

Nurse intervention studies often measure blood pressure, weight, body mass index (BMI), lipids, fasting blood glucose, and HbA1c in terms of physiological outcomes. A significant improvement was observed in BMI, lipids, fasting blood glucose, and HbA1c after a nurse developed and executed support and instructional interventions. In the experimental group, substantial improvements were observed in low-density lipoproteins and HbA1c levels following a telehealth intervention with nurse-directed educational sessions. However, no differences were observed in the case of BMI or blood pressure [42]. In one study, compared to the control group, marked improvements were observed in daily fasting blood glucose levels following a nurse-directed follow-up telephone intervention. A study was also conducted to evaluate the effectiveness of a controlled nursing intervention focused on counselling and education, where a marked amelioration was observed in the HbA1c level in the experimental group compared to the control group. Investigators assessed whether a culturally sensitive, multifaceted, and primary care-based behavioural intervention led by a nurse case manager and/or a community health advisor may result in an improvement in blood pressure, lipids, and HbA1c levels. It was observed that the team involving a community health advisor and nurse was found to be most effective, along with a marked reduction in lipid level and diastolic blood pressure. Although a reduction in HbA1c level was also observed, the data were not significant. A study was developed to lower the cardiovascular risk in individuals with T2DM, where a nurse-led course involved both group and individual sessions to assist participants and discuss self-care domains to develop and meet self-care targets. In this study, both systolic blood pressure and BMI were markedly reduced in the experimental group.

In a randomised controlled trial, a significant improvement was observed in the lipid levels of PWD following algorithm-driven nurse-led telephone care over 20 months. In this study, nurses telephoned patients, evaluated their lipid values, and initiated as well as titrated lipid-lowering medicines. Moreover, participants in the experimental group used fewer healthcare facilities and also experienced improved lipid control [67]. Patients attended a half-day DSME class and were subsequently randomised to either the intervention or usual care group to evaluate the impact of web-based care management on blood pressure and glucose control in individuals with poorly controlled DM. The experimental group received a blood pressure monitor, glucose meter, and a notebook computer and were allowed to view educational modules, upload blood pressure and glucose information, and received a notebook computer. A substantial reduction was observed in systolic blood pressure and HbA1c in individuals in the intervention group. Furthermore, a significant difference was observed between the control and intervention groups. As a substitute to group classes, another study, after providing web-based DSME, observed marked enhancements in HbA1c levels compared to the control group, who participated in conventional DSME classes. Numerous studies have observed marked amelioration in HbA1c level after nursing interventions; however, some studies observed no difference or no change between the control and experimental groups [37,39,45]. Some studies also reported significant improvements in other physiological outcomes for both the control and intervention groups [14,33,68].

### 4.7. Psychosocial Outcomes

Evaluation of QoL and psychosocial assessments such as attitudes towards depression, diabetes, and diabetes-associated anxiety should be continued for PWD. Positive DM outcomes are linked with emotional well-being. Indeed, nursing intervention studies cover psychosocial factors and assess the impact of interventions on these factors. In one study, nurse-led motivational interviewing interventions were carried out to inspire individuals to explore their feelings regarding change and discuss self-management behaviours. Multidisciplinary education was provided to both the control and intervention groups. Both groups also attended a diabetes club that facilitated group discussions of personal experiences of living with DM. After the intervention, a marked amelioration was observed in the QoL in the intervention group in comparison with the control group. Interestingly, both groups experienced markedly lower levels of mean scores for stress, anxiety, and depression, however, there was no noteworthy difference between groups. Exposure of the control group to the diabetes club support and education group might have resulted in the aforementioned lack of difference between the two groups [69]. Compared to the control group, a higher mean score for family domains, social, and psychologic-spiritual was observed in the intervention group in a nurse-led intervention to mediate healthy behaviours in adult PWD. A significantly improved QoL was observed in a study that used a personalised educational and target-setting programme led by a nurse. Significantly lower distress associated with symptoms and markedly higher perceptions regarding QoL were observed in the participants who received a nurse-led diabetes symptom-centred management intervention [14].

## 5. Overcoming the Challenges for Nurses in Diabetes Care

In addition to proper training, it is essential to make fundamental changes in nursing education, health system, policy, and societal levels in order to extend the role of nurses in diabetes care, management, and prevention. These measures are important to ensure that nurses can actually achieve their potential in tackling global challenges. In 2021, a referendum in Switzerland emphasised the significance of nursing and the duty of the country to ensure adequate numbers of nurses to better recognise the importance of the roles of nurses. Despite their significant contributions, nurses are often not sufficiently recognised in governance mechanisms. In Switzerland, the leadership gap in the nursing sector is addressed by involving a cantonal nurse accountable for sharing the unique standpoints of a nurse and communicating with policymakers and various other associates. Altering the kinds of services that they offer and raising the payment for nurse-provided services are effective ways to recognise the importance of the roles of nurses. In a health system, it is important to identify the barriers to the expansion of the roles of nurses. For example, the inability of nurses to prescribe medicines is the sole barrier from a system or legal perspective in Kyrgyzstan, a low-income and middle-income country (LMIC). The perception of the roles of nurses in providing care for non-communicable diseases including diabetes by general people and even doctors is another limitation to the roles of nurses.

It is important to address more practical elements along with the introduction of DSNs to enable nurses to exert roles in prescribing medicines, running nurse-directed diabetes clinics, carrying out diabetes research, and providing diabetes education. A successful extension of nursing roles has been implemented in Thailand, where nurses play various roles as advanced practice nurses, case managers, and educators in diabetes care. Nonetheless, studies have reported that these measures in LMICs resulted in a moderate outcome on DM management associated with reduced blood glucose levels [70]. Nurses ought to have a fundamental role in management and prevention in order to ensure access to diabetes care and to reach global targets. Alongside the global rise in the prevalence of DM, it is important to increase the number of nurses and to improve leadership and training. Moreover, it is crucial to exert fundamental changes within the general nursing environment. It is also important to provide better employment opportunities with clear career paths and better professional recognition. Global issues of migration and retention need customised solutions to ensure that resolving staff shortage issues in one country does not lead to the depletion of nurses in other countries. Along with societal recognition by the population and policymakers, the health system ought to completely acknowledge the significance of the roles played by nurses within the health system.

This acknowledgement should also address the exclusive gender-associated nursing issue, since it remains largely a female profession in the healthcare sector, which is male-dominated in numerous countries. Indeed, nurses require clearly defined responsibilities and roles to provide diabetes care in a health system in order to provide the finest care to the patients they serve. Nurses also require skills and tools to carry out their tasks in different settings. This approach might involve the use or empowerment of diagnostic tools, interprofessional patient education, supportive supervision, task sharing, training on certain disease areas, and prescribing. These roles require recognition in terms of positions and qualifications, along with opportunities for career progression and higher pay. Elevation of an interprofessional method could be an approach to assist such strategies, which includes nursing and medical students learning together during the period of their formative training so that they are well equipped to work as a team in future professional scenarios [71]. Different health professionals should be provided with an environment by the health system that permits them to work as a team for the benefit of PWD [72].

## 6. Conclusions

Nurses closely work with PWD to educate, set self-management goals, and also adopt approaches to assist positive self-management behaviours in the areas of active participation, glucose self-monitoring, medication administration, healthy eating, and physical activity for effective coping and to identify DM complications. Various studies have demonstrated that nursing interventions can result in improved patient outcomes such as QoL, glycaemic control, self-management behaviours, and diabetes knowledge. The summarised findings from numerous studies demonstrate the necessity for continuous nursing interventions in DM management and indicate that more studies are required to establish evidence-based and the most effective interventions

## Figures and Tables

**Figure 1 healthcare-12-00352-f001:**
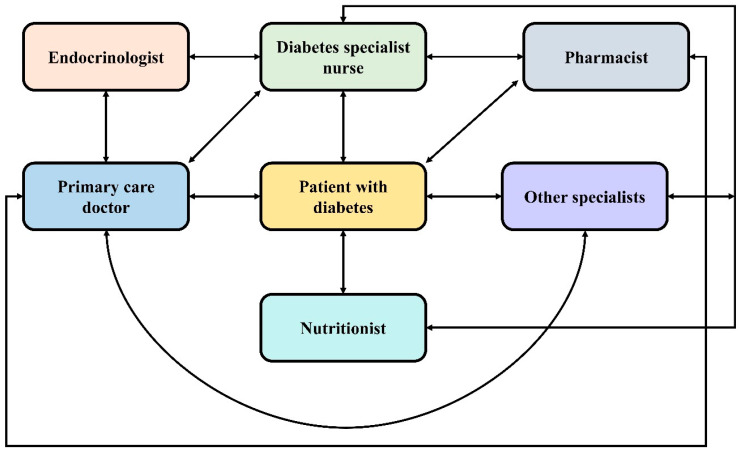
The chronic care model for diabetes mellitus (DM) management. A better collaboration is required between the healthcare providers and specialists for effective DM management.

**Table 1 healthcare-12-00352-t001:** Summary of studies on the impact of nurse-led diabetes self-management education.

Study Goal	Participants	Duration	Study Methods	Study Outcomes	References
Impact of nurse-led diabetes self-management education (DSME) on self-care behaviour and self-care knowledge	360 adult patients with type 2 DM (T2DM)	6 months	Quasi-experimental study design	A statistically significant higher mean score difference was observed in self-care behaviour and self-care knowledge after the provision of DSME	[22,24]
Evaluation of the effect of a nurse-led DSME on glycosylated haemoglobin (HbA1c)	142 patients with T2DM	12 weeks	HbA1c levels, quality of life (QoL), depression, self-management behaviours, lipid profiles, body weight, blood pressure, social support, and alterations in self-efficacy (outcome expectation and efficacy expectation) were measured	Participants in the intervention group exhibited substantial amelioration in diabetes self-management behaviours, outcome expectation, efficacy expectation, body weight, blood pressure, and HbA1c level	[23,25]
Assessment of the effects of nurse-led integrative medicine-based structured education programme	128 patients with T2DM	8 months	Randomised controlled trial	Markedly better self-management performance was observed in specific diet including consumption of vegetables and fruits in patients in the intervention group. The intervention group also showed marked improvements in HbA1c level	[24,26]
Evaluation of the impact of nurse-led DSME on clinical parameters	220 participants	9 months	Before-and-after controlled study with customised fliers and handbooks containing illustrative pictures delivered by trained nurses	Clinically significant effects on both fasting blood sugar and blood pressure were observed in the intervention group; level of HbA1c was markedly decreased win both intervention and comparison groups	[25,27]
Effect of web-based nurse-led education on HbA1c in patients with T2DM	51 participants	12 weeks	A control group pretest–post-test design which reinforced medication adherence, exercise, diet, and self-monitoring of blood glucose levels	Compared to the control group, the intervention group showed better HbA1c control; a substantial change in the percentage of baseline-glycosylated haemoglobin ≥ 7·0% level was observed in the intervention group	[26,28]
Accessibility and impact of nurse-led DSME in rural settings	232 patients with T2DM	56 months	Retrospective cohort study	Substantial improvements were observed at 12 months in participants in psychological distress, QoL, body mass index, and cholesterol; the study also concluded that nurse-led DSME in resource constraints or rural settings can provide highly accessible services to meet the needs	[27,29]
Evaluation of the outcomes of a DSMES programme on patient knowledge of DM and patient medication adherence in an inpatient setting	10 participants	6 months	A descriptive, pre-test–post-test study	After provision of DSMES, a statistically significant amelioration of the patient knowledge of DM was observed, while no significant change was observed in the patients’ medication adherence	[28,30]

## Data Availability

All data are included in this manuscript.

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
