# Peer review of "The Influence of Nurse-Led Interventions on Diseases Management in Patients with Diabetes Mellitus: A Narrative Review"

_healthcare, 2024, doi:10.3390/healthcare12030352_

Round 1

Reviewer 1 Report

Comments and Suggestions for Authors

This is a review of the role of nurses in care and education of patients with diabetes. The author is commended for undertaking a review of an interesting and important topic. It is requested that the manuscript is edited for concise language and presentation of topics. It is suggested that similar topics from different sections be condensed and combined to improve clarity and ‘flow’ of topics. As currently organized, many sections seem redundant. Also, there is ample evidence presented on the effectiveness of nurse-led programs or activities, but could you also elaborate on why nurses are particularly  poised for these roles (as opposed to another diabetes specialist?).

Comments on the Quality of English Language

English language is fine, the manuscript could use some editing for more concise wording.

Reviewer 2 Report

Comments and Suggestions for Authors

Thank you for the opportunity to review your manuscript. The following comments are offered with all due respect to the authors:

While the focus of this paper in the Section 1 Introduction is on the impact of nurse-lead interventions on diabetes management for people with T1DM and T2DM, Section 2 Search Methods ends with a statement about the role of nurses in the prevention of diabetes.  The authors may wish to consider whether the introduction should include information about people with pre-diabetes or if the focus will be on T1DM and T2DM. 

The authors are encouraged to use "T1DM" and "T2DM" consistently throughout the manuscript (i.e., lines 64-65, 92, Table 1).

The authors may wish to review the numbering of sub-sections.  For example, 3.2 is listed twice.  It should be listed once and 3.3, 3.4, 3.5, 3.6, 3.7, and 3.8 should follow.  In Section 4, 4.5 is listed two times.  It should be listed once followed by 4.6 and 4.7.

Table 1 appears to include information from sections 3.1 and 3.2.  If this is correct, the authors may wish to create separate tables for each section or to create a single table with information that corresponds to each subsection listed in order (i.e., 3.1 followed by 3.2, then 3.3).

Figure 1 refers to information from Section 3.2.  It may be helpful to add more context, so the reader clearly understands why the authors highlighted the figure.  For example, it is unclear what "low," "medium," and "high" levels of care entail.  Similarly, references to "normal care" and "usual care" are made. It may help to elaborate by operationalizing these terms.

In the section about "community health advisors," the authors being the section by also using the term "lay health workers," however, they later use the term "promotoras."  The authors may wish to note that different terms are used.

They authors discuss the critical role of nurses as sources of emotional support.  Given the extensive amount of literature on the mental health of people with diabetes and the strategies that can reduce fears and worries, the authors are encouraged to elaborate.  In addition, the authors may wish to include the discussion about communication (lines 293-297) in the section about "interprofessional teamwork."  Lastly, subsection 3.8 includes information about psychological screening (lines 337-348).  The authors may wish to streamline this information in the section on "Psychological care and counselling."

Several important issues are introduced in this paper.  While these are important issues (i.e., gender), this paper may be strengthened by narrowing the focus to ensure that the key topic is adequately addressed. 

Comments on the Quality of English Language

Thank you for the opportunity to review your manuscript.  With regard to the quality of the English language, I would recommend a comprehensive review of grammar and transition phrases.  While not specific to the English language, the authors are encouraged to follow the diabetes language guidelines recommended by Diabetes Australia and the American Diabetes Association.  For example, use "people with diabetes (PWD)" instead of "patient with diabetes (PWD)" and avoid the use of the word "diabetic" as an adjective unless referring to diabetes-related complications.

Reviewer 3 Report

Comments and Suggestions for Authors

Abstract

The abstract is sound. Only DM abbreviation has not been defined before. 

Keywords

I suggest that the author look into arranging the keywords as follows: 

Diabetes mellitus; Chronic patients; Nurse-led interventions; Self-management; Patient outcomes 

Introduction

There are several of statements without reference. It's better to repeat references than to have only one or two references at the end of the paragraph. 

Introduction has covered necessary information. 

Methods

It would be interesting to indicate the Medical Subject heading (MeSH) terms and synonyms in a standardized manner (meaning words written in inverted commas).

Clarify how many people did the search? How was quality assurance assessed to ensure that no study was left out or any repetition? 

The results are well presented, discussion, and conclusion. 

Reviewer 4 Report

Comments and Suggestions for Authors

1.Abstract:There is some repetition in the abstract, particularly in mentioning the various outcomes measured to assess the impact of nursing interventions. Consider streamlining this information for conciseness.

2. While you've defined DM in the introduction, it might be helpful to also define T1DM and T2DM upon first use for readers who might not be familiar with these acronyms. Ensure consistency in terminology. For example, you use "PWD" to refer to people with diabetes, which is good to define, but then use "DISNs" without prior explanation. Consider explaining "DISNs" (Line 60)on its first use as well.Moreover, watch for repetitive phrases. When introducing abbreviations like "DSNs" (assuming it refers to Diabetes Specialist Nurses), consider explicitly stating this upon first use for clarity.For instance, you mention "DM management" multiple times in the same sentence. Consider rephrasing for variety.Ensure consistency in terminology and abbreviations throughout the entire document.

3.Provide a brief rationale for the selection of the specific databases. Why did you choose these databases, and what unique contributions do they offer to the topic?Moreover,you mentioned searching from 2010 to October 2023; why did you choose this Date range?

4.  If you used Boolean operators (AND, OR, NOT) in your search strategy, consider explicitly mentioning them. For example, you could say, "The search utilized combinations of terms such as nursing AND diabetes AND intervention."

5.  Lack of discussion of opposing views: The article focuses on the positive role of nurses in diabetes care, but does not address possible opposing views or controversial issues. Including a discussion of the different views will make the article more comprehensive.

6.Consider adding subheadings to break down the section into more manageable parts. For example, you could have subheadings like "Factors in Self-Care Management," "Patient Education Techniques," and "Impact of Nurse-Led Education."

7.Clarity in Expression:

Line105, "Nevertheless, the efficacy of health education activities is influenced by a patients acceptance of DM," it might be clearer to say "However, the effectiveness of health education activities depends on a patients acceptance of their diabetes."The phrase "healthy coping" might be more understandable if expanded upon briefly.

Line106, "Moreover, various sociodemographic factors including an individuals education level can also affect adherence to self-care in DM management," consider rephrasing for smoother flow, e.g., "Additionally, sociodemographic factors, such as an individuals education level, can impact adherence to self-care in DM management."

Line 126, "HbA1c test is commonly carried out to measure the average blood glucose levels in the past 3 months," you might consider adding a comma after "HbA1c test" for better readability.

Comments on the Quality of English Language

none

Round 2

Reviewer 1 Report

Comments and Suggestions for Authors

Thank you for the opportunity to review this manuscript. Generally, it seems the manuscript could still be shortened, as there seems to be redundancy between section 3 and 4. Section 3 can be edited to only describe various roles and section 4 describes outcomes. This would enhance clarity of the manuscript.

Specific Comments:

Consider editing title to “The influence of nurse-led interventions on disease management in patients with diabetes mellitus

Line 56: remove ‘more’

Line 60: physicians and doctors seem redundant.

Line 130-132 is confusing- in that study the ‘usual care’ received more than usual care? Can you please clarify?

136- study by Raballo et al. could you specify if the group care was led by nurses, or how that plays into the “role of nurses in management of diabetes”?

196-198- mentioned other studies that assess novel educational methods, what were the outcomes found?

230-231- it may be helpful to provide some examples for the training/regulatory requirements to allow nurses to write prescriptions. This is mentioned again in section 3.8, I think it would be helpful to combine these sections.

 335-336- sentence meaning is not clear, “APNs particularly associated with PWD”. Please clarify.

Author Response

file attached
